# When a Red Herring is Not a Red Herring: Using Compositional Methods to Improve the Detection of Non-Compositional Phrases

## Abstract

Non-compositional phrases such as *red herring* and weakly compositional phrases such as *spelling bee* are an integral part of natural language. They are also the phrases that are difficult, or even impossible, for good compositional distributional models of semantics. Compositionality detection therefore provides a good testbed for compositional methods. We compare an integrated compositional distributional approach, using sparse high dimensional representations, with the ad-hoc compositional approach of applying simple composition operations to state-of-the-art neural embeddings.

## 1 Introduction

In recent years, distributional representations of words have received a lot of interest. In many applications, the ability to cluster or find similar words in terms of their distribution in text or their hypothesised semantic similarity has a massive potential to reduce the sparse data problem. Early research in the field (Hindle, 1990; Grefenstette, 1994; Lin, 1998; Lee, 1999; Curran, 2004; Weeds and Weir, 2005; Padó and Lapata, 2007) investigated distributional representations which were built directly from corpus co-occurrence counts; such representations are now commonly referred to as count or explicit vector representations. The models considered varied in terms of the use of different association functions (Curran, 2004), the use of different similarity measures (Lee, 1999; Weeds and Weir, 2005) and whether to define context in terms of proximity or grammatical dependencies (Padó and Lapata, 2007). More recently the trend has been towards using neural models (Mikolov et al., 2013;

Pennington et al., 2014) to create dense, low dimensional representations, commonly referred to as word embeddings, built by training the models to predict corpus co-occurrences. As has been noted elsewhere (Pennington et al., 2014), whilst appearing at first sight very different, both count-based methods and prediction-based methods have in common the fact that they probe the underlying co-occurrence statistics in the corpus. In fact, Levy and Goldberg (2014) demonstrated that the skip-gram model with negative sampling (SGNS) proposed by Mikolov et al. (2013) is an implicit factorisation of the positive pointwise mutual information (PPMI) matrix commonly used in count-based methods.

One current focus within the field of distributional semantics is enabling systems to make inferences about phrase-level or sentence-level similarity. One approach (Turney, 2012) is to model the similarity of two phrases or sentences as a function of word-level similarities. An alternative approach (Mitchell and Lapata, 2010) is to build phrase or sentence-level representations by composing word-level representations and then measuring similarity directly. However, whilst this second approach has proved popular, success, usually measured in terms of correlation with human similarity judgments, has been limited. For example, Dinu et al. (2013) reported results for a number of state-of-the art compositional methods on a number of phrase-based benchmark tasks. In those experiments, correlation with human judgements for intransitive sentences (Mitchell and Lapata, 2008) does not exceed 0.30.

However, evaluating measures of phrase-level similarity directly against human judgments of similarity ignores the problem that it is not always possible to determine meaning in a compositional manner. If we compose the meaning representations for *red* and *herring*, we would ex-

pect to get a very different representation from the one which could be directly inferred from corpus observations of the phrase *red herring*. This would undoubtedly have an impact on our similarity measurements with other words and phrases. Whilst it should be possible to construct evaluation datasets which avoid clearly idiomatic phrases such as *red herring*, non-compositionality is an integral part of language which should not be ignored (Sag et al., 2002). Further, McCarthy et al. (2003) noted that the compositionality of a phrase should not be judged categorically, rather it should be viewed as existing on a continuum or a scale. Thus any judgements of the similarity of two composed phrases are confounded by the degree to which those phrases are compositional.

Reddy et al. (2011) introduced a new dataset of 90 compound nouns together with human judgments of compositionality. Using this benchmark, compositional methods can be evaluated by correlating the similarity of composed and observed (or holistic) phrase representations with human judgments of compositionality.

In this paper, we use this dataset to investigate the extent to which the underlying definition of context has an effect on a model's ability to support composition. In particular, we distinguish between typed and untyped contextual features. For example, in traditional count-based models, contextual features based on proximity are usually untyped whereas contextual features based on dependency relations may be typed (i.e., include the name of the dependency relation) or untyped (Baroni and Lenci, 2010). Further, following Weeds et al. (2014) we investigate the use of higher order dependency paths in contextual features in order to support composition in typed vector space models. We compare these models, where composition is an integral part of the distributional model, with the commonly employed approach of applying naïve compositional operations to state-of-the-art distributional representations.

## 2   Related Work

In distributional semantic models, composition has typically been tackled by taking vector representations for words (Turney and Pantel, 2010) and combining them using some function to produce a data structure that represents the phrase or sentence. The simplest functions which can be used in composition are pointwise addition and multiplication. From a linguistic perspective, these are naïve since they are commutative, completely ignoring the order and structure of the phrase or sentence. However, Mitchell and Lapata (2008, 2010) found that additive and multiplicative functions applied to proximity-based vector representations were no less effective than more complex functions when performance was assessed against human similarity judgements of simple paired phrases. More recently, Berant and Liang (2014) achieved state-of-the-art results in question-answering where the paraphrase model included a vector space model component. This vector space model component carried out composition by adding neural word embeddings obtained with the word2vec tool (Mikolov et al., 2013).

Over the past 6 years, other more linguistically motivated models of composition have been proposed e.g., the full additive model (Guevara, 2010), the lexical function method (Baroni and Zamparelli, 2010), the full lexical model (Socher et al., 2012) and various tensor methods (Coecke et al., 2011; Grefenstette et al., 2013). These methods all share the idea, taken from formal semantics, of function application derived from syntactic structure. In an evaluation across 3 different benchmark tasks (Dinu et al., 2013), the lexical function model was shown to be consistently the best-performing. However, in the composition of adjective-noun phrases, the simple additive and multiplicative models were still shown to be highly competitive.

Milajevs et al. (2014) compared neural word representations with count-based vectors in a variety of tasks using a variety of naïve and tensor-based compositional models. Across 4 different tasks (word sense disambiguation, sentence similarity, paraphrase detection and dialogue act tagging), the neural word representations consistently outperformed the traditional count-based vectors. However, as concluded by the authors, this may well be due to differences in the size and nature of the corpora from which the different representations were obtained. Considering the results for the neural word representations, pointwise addition outperformed all of the other compositional models considered on 3 out of the 4 tasks. Tensor-based composition performed better than pointwise addition on just the verb disambiguation task, where the authors argue that verb senses depend strongly on the arguments of the verb.

Hashimoto et al. (2014) integrated a variety of syntactic and semantic dependencies into neural models in order to jointly learn composition functions and word representations. Whilst these models are well-motivated and achieved some state-of-the art results on the Mitchell and Lapata (2010) phrase similarity task, the baseline of adding standard neural word embeddings produced by the word2vec tool proved particularly hard to beat.

Hermann et al. (2012) proposed using generative models for modeling the compositionality of noun-noun compounds. Using interpolation to mitigate the sparse data problem, their model beat the baseline of weighted addition on the Reddy et al. (2011) evaluation task when trained on the BNC. However, these results were still siginificantly lower than those reported by Reddy et al. (2011) using the larger ukWaC corpus.

## 3 Composition via Anchored Packed Trees

Elsewhere (reference suppressed), we proposed a distributional compositional approach where distributional features are based on anchored packed trees (APTs). The essence of the APT approach is that distributional features should encode complete dependency paths from the target word to each context word and that these representations should be properly aligned before composition.

| token | 1st order | 2nd order |
|---|---|---|
| grad/N | $\langle\overline{\text{NMOD}}, student\rangle$ | $\langle\overline{\text{NMOD}}.\overline{\text{NSUBJ}}, fold\rangle$ |
| student/N | $\langle\text{NMOD}, grad\rangle$, $\langle\overline{\text{NSUBJ}}, fold\rangle$ | $\langle\overline{\text{NSUBJ}}.\text{DOBJ}, clothes\rangle$ |
| fold/V | $\langle\text{NSUBJ}, student\rangle$, $\langle\text{DOBJ}, clothes\rangle$ | $\langle\text{NSUBJ}.\text{AMOD}, grad\rangle$, $\langle\text{DOBJ}.\text{NMOD}, dry\rangle$ |
| dry/J | $\langle\overline{\text{AMOD}}, clothes\rangle$ | $\langle\overline{\text{AMOD}}.\overline{\text{DOBJ}}, fold\rangle$, $\langle\overline{\text{AMOD}}.\text{AMOD}, clean\rangle$ |
| clothes/N | $\langle\text{AMOD}, dry\rangle$, $\langle\overline{\text{DOBJ}}, fold\rangle$ | $\langle\overline{\text{DOBJ}}.\text{NSUBJ}, student\rangle$ |

Table 1: Typed higher order distributional features

For example, consider the sentence *"The grad student folded the dry clothes."* Table 1 shows some of the features that this would generate in an APT lexicon for each of the content tokens[1]. Features are typed using the complete dependency

---

[1] We assume lemmatisation in these examples since in practice it reduces sparsity, but it is not necessary from a theoretical point of view. In these examples, we also omit the POS tags on the context words since they can be easily inferred by the reader.

path from the token to the context word. The inverse of dependency relation $R$ is denoted by $\overline{R}$. The length of the dependency path is referred to as the order of the distributional feature. For compactness, we only show features up to order 2.

From Table 1, it is clear, as noted by Weeds et al. (2014), that features of words with different parts of speech do not immediately sit in the same space. This is because words of different parts of speech play different roles in syntactic (and the associated semantic) relations. In this example, all of the observed first order features of adjectives are of type $\overline{\text{AMOD}}$ whereas the nouns have observed first order features of types AMOD, NMOD, $\overline{\text{NMOD}}$, $\overline{\text{NSUBJ}}$ and $\overline{\text{DOBJ}}$. The use of an intersective pointwise composition operation such as `multiply` would largely lead to zero vectors, which is obviously not the desired result of composition. A pointwise composition operation which performs a union of the features such as `add` would largely lead to a concatenation of the two vectors.

However, the second observation that we make based on Table 1 is that the 2nd order features of, say, adjectives, correspond to the 1st order features of nouns. Similarly, whilst not shown, the 3rd order features correspond to the 2nd order features and the 4th order features correspond to the 3rd order features. The only difference is that the features in the adjective space have the prefix $\overline{\text{AMOD}}$ which is the path which is traversed between the adjective and the noun in the parse tree.

Accordingly, to construct a vector for a phrase from the perspective of its head (i.e., so that it has features in the same type space as its head), we first offset all of the dependent vectors in accordance with $\delta$, the path to the dependent from the head in the phrase or sentence being composed. During offsetting each feature type is prepended by the given path and reduction applied. The reduced co-occurrence type produced from $\tau$ is denoted $\downarrow(\tau)$, and defined as follows:

$$\downarrow(\tau) = \begin{cases} \downarrow(\tau_1\tau_2) & \text{if } \tau = \tau_1\, r\, \bar{r}\, \tau_2 \text{ or } \tau = \tau_1\, \bar{r}\, r\, \tau_2 \\ & \text{for some } r \in R \\ \tau & \text{otherwise} \end{cases}$$

(1)

Reduction essentially means that a relation cancels with its inverse relation i.e., the occurrence of a relation adjacent to its inverse relation will be replaced by the empty relation. Note, as discussed in (reference suppressed), reduction does introduce

| $\delta$ | *student*/N | *clothes*/N |
|---|---|---|
| $\epsilon$ | $\langle$NMOD, $grad\rangle$, $\langle\overline{\text{NSUBJ}}, fold\rangle$, $\langle\overline{\text{NSUBJ}}.\text{DOBJ}, clothes\rangle$ | $\langle$AMOD, $dry\rangle$, $\langle\overline{\text{DOBJ}}, fold\rangle$, $\langle\overline{\text{DOBJ}}.\text{NSUBJ}, student\rangle$ |
| NSUBJ | $\langle\epsilon, fold\rangle$, $\langle$DOBJ, $clothes\rangle$, $\langle$NSUBJ.NMOD, $grad\rangle$ | $\langle$NSUBJ.AMOD, $dry\rangle$ |
| NMOD | $\langle$NMOD.NMOD, $grad\rangle$, $\langle$NMOD.$\overline{\text{NSUBJ}}, fold\rangle$ | $\langle$NMOD.AMOD, $dry\rangle$ |

Table 2: Features of *student*/N and *clothes*/N given different offsets $\delta$.

zero'th order features (with type $\epsilon$) in both elementary and offset representations. Further, all co-occurrence types are required to have a tree-based interpretation ($\tau \in \overline{R}^* R^*$) which leads to the elimination of some incompatible co-occurrence types. Table 2 shows the result of offsetting the features of *student*/N and *clothes*/N in Table 1 with three different paths 1) $\epsilon$, 2) NSUBJ and 3) NMOD as required in the compositions of the phrases *lazy student*, *student submits* and *student halls* respectively.

Having aligned the vector spaces, it is possible to carry out any pointwise composition operation, such as `add` or `multiply`. The overall result will be sensitive to the structure in the composed phrase because different structures lead to different alignments.

However, the focus of the current study is noun compounds, the majority of which are generally tagged and parsed as noun-noun compounds (e.g., *grad student*/NN). In this particular instance, it is less apparent that the vector spaces need to be aligned before composition. There will be other instances of the word *grad*/N where it is used as the object or subject of verbs. These contexts may well be good indicators of what make good contexts for *grad student*/N (particularly in this example since *grad*/N and *grad student*/NN are often used synonymously). Therefore, we also consider the typed dependency model where features are higher order dependency paths but alignment is not carried out before composition.

## 4 Compositionality of compound nouns

Compositionality detection, as described in Reddy et al. (2011), involves deciding whether a given multiword expression is compositional or not i.e., whether the meaning can be understood from the literal (simplex) meaning of its parts. Reddy et al. (2011) introduced a dataset consisting of 90 compound nouns along with human judgments of their literality or compositionally at both the constituent and the phrase level. All judgments are given on a scale of 0 to 5, where 5 is high. For example, the phrase *climate change* is deemed to take its meaning literally from both constituents and also deemed to be a literal phrase. Conversely, the phrase *gravy train* has low scores for the literalness of the phrase and of the use of each constituent within the phrase. The phrase *cocktail dress* is deemed to be literal in its use of the second constituent but not the first whereas the phrase *spelling bee* is deemed to have high literalness in its use of the first constituent but not in its use of the second. Both of these examples are considered to have a medium level of literalness with respect to the whole phrase.

Reddy et al. (2011) further investigated a number of ways of detecting compositionality using vector based models of word meaning. They experimented with both constituent based models and compositionality function based models. In constituent based models, the compositionality of the phrase is considered to be a function of the similarity of each of the constituent's vectors to the observed phrase vector. This is based on the intuition that if a constituent is used literally within a phrase then it is highly likely that the compound and the constituent share co-occurrences. In compositionality function based methods, the compositionality of the phrase is determined by first composing the constituents using some function and then measuring the similarity of the composed vector to the observed vector. The intuition here is that a good compositionality function is highly likely to return vectors which are similar to observed phrasal vectors for compositional phrases but much less likely to return similar vectors for non-compositional phrases. Reddy et al. (2011) carried out experiments with a vector space model built from ukWaC (Ferraresi et al., 2008) using untyped co-occurrences (window size=100). They used 3-fold cross-validation to estimate model parameters and found that using weighted addition outperformed multiplication as a compositionality function and that this also outperformed all of the constituent based models. With the optimal settings in their experiments they achieved a Spearman's rank correlation coefficient of 0.714 with the human judgments.

We adapt the experiment described above to

look at the effectiveness of APT composition in predicting the first-order dependency features of compound nouns. When a phrase is compositional, we expect that the second- (and zero-) order features of the modifier composed with the first-order features of the head noun will be a good predictor of the first-order features of the compound noun. For example, we expect the second-order dependency features of *spelling*, which would include evidence from other uses of *spelling* as a modifier (e.g. *spelling test*) to be more indicative of the co-occurrences of *spelling bee* than the first-order dependency features of *spelling*.

## 5 Experimental set-up

For consistency with the experiments of Reddy et al. (2011), the corpus used in this experiment is the same fully-annotated version of the web-derived ukWaC corpus (Ferraresi et al., 2008). As described in Grefenstette et al. (2013), this corpus has been tokenised, POS-tagged and lemmatised with TreeTagger (Schmid, 1994) and dependency-parsed with the Malt Parser (Nivre, 2004). It contains about 1.9 billion tokens.

Before creating our APT lexicon and word embeddings from the dependency-parsed corpus, we further preprocessed the corpus by identifying occurrences of the 90 target noun phrases and recombining them into a single lexical item. Note that whilst all of the phrases are considered to be compound nouns, there are a number of possible dependency relationships which can occur between them. In the corpus we identified $1,236,264$ occurrences of the candidate lemmas occurring contiguously (in the correct order). 76% of these had some kind of dependency relationship where the second lemma was the head. We note that the majority of occurrences where no dependency relationship was observed, this was due to the parser incorrectly parsing a three noun compound phrase, e.g., parsing the phrase *interest rate rise* so that both *interest* and *rate* modify *rise*. Of the occurrences where some kind of dependency relationship in the correct direction was identified, 95% of were an NMOD relationship. However, the modifier in an NMOD relationship can be an adjective or a noun. Many compounds (e.g. *graduate student* and *silver screen*) are seen with the modifier tagged both as a noun and as an adjective. So that we can carry out composition of the correct tokens (e.g. *graduate* as an adjective as opposed to *grad-*

*uate* as a noun) according to observed dependency relations, the token for the compound records both the individual lemmas and the dependency relation observed between them in the corpus. Where a compound is seen with multiple dependency relationships we selected the dependency relationship which occurred most frequently.

During this preprocessing stage, other dependency paths including the head constituent of the phrase are modified to include the new compound token. In this experiment, dependency paths including the modifying constituent are ignored. The rationale for this is that if the modifier is being modified e.g. as in *(recently graduated) student*, then this is not a modification which can be applied to the phrase — we do not apply adverbs to compound nouns — rather it is a modification of an internal part of the phrase.

Having preprocessed the corpus to contain compound nouns, we created elementary representations for every token in the corpus. Note that the elementary representation for the constituent of a compound phrase will not contain any of the contextual features associated with the compound phrase token unless they occurred with the constituent in some other context. If we allowed the same single observed contextual feature to feed into the representation of the compound and of the constituents, then intersective methods of composition would do very well at recreating the observed vector — ignoring any feature selection, recall would be 100%. Following Weeds et al. (2014) we expect a good method of composition to be able to infer the representation of a phrase without ever having observed the phrase in the corpus.

We will now discuss the parameters we have explored in terms of the construction and composition of elementary representations in each model.

### 5.1 APT model

In relation to the construction of the elementary APTs, the most obvious parameter is the nature of the weight associated with each feature. We consider both the use of normalised counts and PPMI values. If $w$ is a target word, $w'$ a context word and $\tau$ a dependency path from $w$ to $w'$ then the normalised count is simply:

$$p(w', \tau \,|\, w) = \frac{\#\langle w, \tau, w' \rangle}{\#\langle w, *, * \rangle} \qquad (2)$$

Levy et al. (2015) showed that the use of context distribution smoothing ($\alpha = 0.75$) in the PMI cal-

culation can lead to performance comparable with state-of-the-art word embeddings on word similarity tasks. We use this modified definition of PMI and experiment with $\alpha = 0.75$ and $\alpha = 1$

$$\text{PMI}\left(w, w'; \tau, \alpha\right) = \frac{\#\langle w, \tau, w'\rangle \#\langle *, \tau, *\rangle^\alpha}{\#\langle w, \tau, *\rangle \#\langle *, \tau, w'\rangle^\alpha} \quad (3)$$

We also carried out some experiments with shifted PMI which is analogous to the use of negative sampling in word embeddings (Levy et al., 2015). However, in this study we found that shifting PMI tended to have a strong negative effect on results. We suspect this is due to the low frequency of many of the compound nouns in the corpus. Removing features which tend to go with lots of things (low positive PMI) means that these phrases appear to have been observed in a very small number of (highly informative) contexts. If the composition process fails to recover one or more of these contexts, it has a very big impact on similarity. For compactness, we do not include the results with shifted PMI here.

Having constructed elementary APTs[2], the APT composition process involves aligning those elementary APTs and composing the associated vectors of weights. However, the composition operation used after alignment is not fixed. Here, we have investigated using $\bigsqcup_{\text{INT}}$, which takes the minimum of each of the constituent's feature values and $\bigsqcup_{\text{UNI}}$, which performs pointwise addition on the aligned vector spaces. Following Reddy et al. (2011), who found that weighted addition worked best in their experiments, when using the $\bigsqcup_{\text{UNI}}$ operation, we have experimented with weighting the contributions of each constituent to the composed APT representation using the parameter, $h$. For example, if $\mathbf{A}_2$ is the APT associated with the head of the phrase and $\mathbf{A}_1^\delta$ is the properly aligned APT associated with the modifier where $\delta$ is the dependency path from the head to the modifier (e.g. NMOD or AMOD), the composition operations can be defined as:

$$\bigsqcup_{\text{INT}}\left\{\mathbf{A}_1^\delta, \mathbf{A}_2\right\} \quad (4)$$

$$\bigsqcup_{\text{UNI}}\left\{(1-h)\mathbf{A}_1^\delta, h\mathbf{A}_2\right\} \quad (5)$$

---

[2] The size of the feature space for nouns is approximately $80,000$ dimensions (when including only first-order paths) and approximately $230,000$ dimensions (when including paths up to order 2).

We have also considered composition without alignment of the modifier's APT, i.e, using $\mathbf{A}_1$:

$$\bigsqcup_{\text{INT}}\left\{\mathbf{A}_1, \mathbf{A}_2\right\} \quad (6)$$

$$\bigsqcup_{\text{UNI}}\left\{(1-h)\mathbf{A}_1, h\mathbf{A}_2\right\} \quad (7)$$

In general, we would expect there to be little overlap between APTs which have not been properly aligned. However, in the case where $\delta$ is the NMOD relation, i.e., the internal relation in the vast majority of the compound phrases, there may well be considerable overlap between the conventional first-order dependency features of the modifier and the head. In order to examine the contribution of both the aligned and unaligned APTs in the composition process, we used a hybrid method where the composed representation is defined as:

$$\bigsqcup_{\text{INT}}\left\{(q\mathbf{A}_1^\delta + (1-q)\mathbf{A}_1), \mathbf{A}_2\right\} \quad (8)$$

$$\bigsqcup_{\text{UNI}}\left\{(1-h)(q\mathbf{A}_1^\delta + (1-q)\mathbf{A}_1), h\mathbf{A}_2\right\} \quad (9)$$

In the case where representations consist of APT weights which are normalised counts, PPMI is estimated after composition. Similarity between composed and observed phrasal vectors (restricted to 1st order dependency features[3]) is then computed using the cosine measure.

## 5.2 Neural word embeddings

For each word and compound phrase, neural representations were constructed using the word2vec tool (Mikolov et al., 2013). Whilst it is not possible or appropriate to carry out an exhaustive parameter search, we experimented with a number of commonly used and recommended parameter settings. In particular, we investigate both the `cbow` and `skip-gram` models with 50, 100 and 300 dimensions. We also experiment with the subsampling threshold, trying $10^{-3}$, $10^{-4}$ and $10^{-5}$. As recommended in the documentation, in the `cbow` model we use a window size of 5 and and in the `skip-gram` model we use a window size of 10.

---

[3] The rationale for this is that, whilst both composed and observed representations contain higher-order dependency features, the second-order (and third-order) paths in the composed representations are not reliable since the elementary APTs only contained paths up to order 2.

| Compositional Model | PPMI $\alpha = 1$ | | PPMI $\alpha = 0.75$ | |
|---|---|---|---|---|
| | CF | CS | CF | CS |
| Aligned $\sqcup_{\text{INT}}$ (Eq. 4) | 0.72 | 0.70 | **0.75** | 0.72 |
| Aligned $\sqcup_{\text{UNI}}$ (Eq. 5) | 0.71 | 0.72 | 0.72 | **0.75** |
| Unaligned $\sqcup_{\text{INT}}$ (Eq. 6) | 0.74 | 0.72 | 0.72 | 0.73 |
| Unaligned $\sqcup_{\text{UNI}}$ (Eq. 7) | 0.77 | 0.75 | **0.78** | 0.77 |
| Hybrid $\sqcup_{\text{INT}}$ (Eq. 8) | 0.74 | 0.73 | 0.73 | 0.73 |
| Hybrid $\sqcup_{\text{UNI}}$ (Eq. 9) | 0.78 | 0.78 | **0.79** | 0.76 |

Table 3: Average $\rho$ between human judgements and phrase compositionality scores using APT representations.

Early experiments with different composition operations, showed `add` to be the only promising option. Consequently, all of the results reported here are for weighted addition. Similarity between composed and observed representations is computed using the cosine measure.

## 6 Results

In order to estimate the model parameters $h$ and $q$, we use repeated 3-fold cross-validation. Optimum values of the parameters are selected using the training samples. Results are reported in terms of average Spearman rank correlation scores ($\rho$) of phrase compositionality scores with human judgements on the corresponding testing samples. For clarity, we do not include the errors in the tables, but we have used a sufficiently large number of repetitions that these are all small ($\leq 0.0015$) and thus any difference observed which is greater than 0.005 is statistically significant at the 95% level. Boldface is used to indicate the best performing configuration of parameters for a particular compositional model.

Table 3 summarises results for different composition operations and parameter settings using APT representations. We see that all of the results using standard PPMI ($\alpha = 1$) and smoothed PPMI ($\alpha = 0.75$) significantly outperform the result reported in Reddy et al. (2011), which used an untyped dependency space. Smoothing the PPMI calcu-

| | $t = 10^{-3}$ | $t = 10^{-4}$ | $t = 10^{-5}$ |
|---|---|---|---|
| `cbow`, 50d | 0.73 | 0.65 | 0.62 |
| `cbow`, 100d | **0.74** | 0.65 | 0.64 |
| `cbow`, 300d | 0.70 | 0.70 | 0.67 |
| `skip-gram`, 50d | 0.59 | 0.64 | 0.62 |
| `skip-gram`, 100d | 0.62 | 0.64 | 0.64 |
| `skip-gram`, 300d | 0.63 | 0.64 | **0.68** |

Table 4: Average $\rho$ between human judgements and phrase compositionality scores using neural word embeddings

lation with a value of $\alpha = 0.75$ generally has a small positive effect. On average, the results when normalised counts are composed and PPMI is calculated as part of the similarity calculation (CF) are slightly higher than the results when PPMI weights are composed (CS) . However, the differences are small and it is possible that different parameter settings would lead to better results for CS. In general, the unaligned model outperforms the aligned model. However, a small but significant performance gain is generally made using the hybrid model. This suggests that aligned APT composition and unaligned APT composition are predicting different contexts for compound nouns which all contribute to forming a better estimate of the compositionality of the phrase. Regarding different composition operations, $\sqcup_{\text{UNI}}$ generally outperforms $\sqcup_{\text{INT}}$. However, it would appear that it may be better to use the intersective operation when composing aligned APTs where the weights are normalised counts.

Table 4 summarises results for different parameter settings for the neural word embeddings. Looking at the results in Table 4, we see that the `cbow` model significantly outperforms the `skip-gram` model. Using the `cbow` model with 100 dimensions and a subsampling threshold of $t = 10^{-3}$ gives a performance of 0.74 which is significantly higher than the previous state-of-the-art reported in Reddy et al. (2011). Since both of these models are based on untyped co-occurrences, this performance gain can be seen as the result of parameter optimisation.

We note that the `cbow` model seems to benefit from a higher subsampling threshold than recommend elsewhere in the literature (Levy et al., 2015). Since subsampling with lower thresholds is analogous to shifted PPMI, we hypothesise this

| Model | Average $\rho$ |
|---|---|
| RAND | 0.02 |
| FREQ | 0.63 |
| REDDY | 0.71 |
| skip-gram | 0.68 |
| cbow | 0.74 |
| aligned APT | 0.75 |
| unaligned APT | 0.78 |
| hybrid APT | **0.79** |

Table 5: Summary of optimal $\rho$ values between phrase compositionality scores and human judgements for each compositional model

'optimisation' was not beneficial in the current task for the same reason. Many of the compound phrases are comparatively low frequency tokens in the corpus and therefore subsampling their co-occurrences with high frequency words damages their representations. We also tried using lower subsampling thresholds with the 50-dimensional and 100-dimensional cbow model, but results did not improve.

Finally, we note that consistently across all applicable models, optimal values of both $h$ and $q$ were in the range $[0.3, 0.5]$. This suggests that for this dataset the modifier is slightly more informative than the head in determining the compositionality of compound nouns. It also suggests that the unaligned typed dependency features are slightly more informative than the aligned typed dependency features.

Table 5 summarises the results across all of the compositional models. We also report three baselines. The RAND baseline assigns a random compositionality score to a compound. The FREQ baseline uses the observed frequency of the compound in the corpus as a predictor of compositionality. The REDDY baseline is the reported state-of-the-art result (Reddy et al., 2011) which uses weighted addition of untyped co-occurrence vectors obtained from the ukWaC corpus.

Looking at the results in Table 5, we first note that the FREQ baseline performs much better than the RAND baseline. For this dataset, there is a significant amount of correlation between frequency and human judgments of compositionality i.e. the more frequently occurring compounds tend to be compositional. However, we see that all of the methods for predicting compositionality outperform this baseline. Further the methods based on APT representations outperform the methods based on neural word embeddings. This suggests that the dependency paths encoded in the typed contextual features are highly informative in the task of determining the compositionality of a noun compound. When a linear combination of aligned and unaligned APTs is allowed, i.e. using the hybrid method, optimal performance is achieved.

## 7 Conclusions and Further Work

We have demonstrated a number of ways in which compositionality detection for compound nouns can be improved. First, combining traditional compositional methods with state-of-the-art low-dimensional word representations significantly improves results. However, further improvements can be achieved using an integrated compositional distributional approach. This approach maintains syntactic structure within the contextual features of words which is then central to the compositional process. We argue that some knowledge of syntactic structure is crucial in the fine-grained understanding of language. Since compositionality detection also provides a way of evaluating compositional methods without confounding judgements of phrase similarity with judgements of compositionality, it appears that the APT approach to compositionality is reasonably promising. Further work is of course needed with other datasets and other types of phrase.

The use of compositional methods to detect compositionality may also lead to improved compositional methods. For example, one reason why intersective approaches to composition may work less well than additive approaches is the sparse nature of the elementary representations. Improvements can be made (reference suppressed) by carrying out distributional smoothing (Dagan et al., 1993) on the elementary representations. However, further improvements might be made by first detecting non-compositional phrases. If a phrase is judged to be non-compositional, i.e., its composed representation is too dissimilar from its observed representation, the observed phrasal representation should be smoothed rather than its constituents' representations.

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
