# Peer review of "When a Red Herring is Not a Red Herring: Using Compositional Methods to Improve the Detection of Non-Compositional Phrases"

_CoNLL 2016 — decision unknown_

[Official Review · Reviewer 1 · rating 3 · confidence 4]
soundness 4 · originality 2 · clarity 3 · impact 2 · substance 2 · appropriateness 5 · meaningful comparison 4 · replicability 4 · presentation format Poster

General comments
=============================
The paper reports experiments on predicting the level of compositionality of
compounds in English. 
The dataset used is a previously existing set of 90 compounds, whose
compositionality was ranked from 1 to 5
(by a non specified number of judges).
The general form of each experiment is to compute a cosine similarity between
the vector of the compound (treated as one token) and a composition of the
vectors of the components.
Evaluation is performed using a Spearman correlation between the cosine
similarity and the human judgments.

The experiments vary
- for the vectors used: neural embeddings versus syntactic-context count
vectors
- and for the latter case, whether plain or "aligned" vectors should be used,
for the dependent component of the compound. The alignment tries to capture a
shift from the dependent to the head. Alignment were proposed in a previous
suppressed reference.

The results indicate that syntactic-context count vectors outperform
embeddings, and the use of aligned alone performs less well than non-modified
vectors, and a highly-tuned combination of aligned and unaligned vectors
provides a slight improvement.

Regarding the form of the paper, I found the introduction quite well written,
but other parts (like section 5.1) are difficult to read, although the
underlying notions are not very complicated. Rephrasing with running examples
could help.

Regarding the substance, I have several concerns:

- the innovation with respect to Reddy et al. seems to be the use of the
aligned vectors
but they have been published in a previous "suppressed reference" by the
authors.

- the dataset is small, and not enough described. In particular, ranges of
frequences are quite likely to impact the results. 
Since the improvements using aligned vectors are marginal, over a small
dataset, in which it is unclear how the choice of the compounds was performed,
I find that the findings in the paper are quite fragile.

More detailed comments/questions
================================

Section 3

I don't understand the need for the new name "packed anchored tree".
It seems to me a plain extraction of the paths between two lexical items in a
dependency tree,
namely a plain extension of what is traditionally done in syntactic
distributional representations of words
(which typically (as far as Lin 98) use paths of length one, or length 2, with
collapsed prepositions).

Further, why is it called a tree? what are "elementary APTs" (section 5.1) ?

Table 2 : didn't you forget to mention that you discard features of order more
than 3 
(and that's why for instance NMOD.overline(NSUBJ).DOBJ does not appear in
leftmost bottom cell of table 2
Or does it have to do with the elimination of some incompatible types you
mention
(for which an example should be provided, I did not find it very clear).

Section 4:

Since the Reddy et al. dataset is central to your work, it seems necessary to
explain how the 90 compounds were selected. What are the frequency ranges of
the compounds / the components etc... ? There is a lot of chance that results
vary depending on the frequency ranges.

How many judgments were provided for a given compound? Are there many compounds
with same final compositionality score? Isn't it a problem when ranking them to
compute the Spearman correlation ?

Apparently you use "constituent" for a component of the N N sequence. I would
suggest "component", as "constituent" also has the sense of "phrase" (syntagm).

"... the intuition that if a constituent is used literally within a phrase then
it is highly likely that the compound and the constituent share co-occurrences"
: note the intuition is certainly true if the constituent is the head of the
phrase, otherwise much less true (e.g. "spelling bee" does not have the
distribution of "spelling").

Section 5

"Note that the elementary representation for the constituent of a compound
phrase will not contain any of the contextual features associated with the
compound phrase token unless they occurred with the constituent in some other
context. "
Please provide a running example in order to help the reader follow which
object you're talking about.
Does "compound phrase token" refer to the merged components of the compound?

Section 5.1

I guess that "elementary APTs" are a triplet target word w + dependency path r
+ other word w'?
I find the name confusing.

Clarify whether "shifted PMI" refer to PMI as defined in equation (3).

"Removing features which tend to go with lots of
 things (low positive PMI) means that these phrases
 appear to have been observed in a very small num-
 ber of (highly informative) contexts."
Do "these phrases" co-refer with "things" here?
The whole sentence seems contradictory, please clarify.

"In general, we would expect there to be little 558
overlap between APTs which have not been prop-
erly aligned."
What does "not properly aligned" means? You mean not aligned at all?

I don't understand paragraph 558 to 563.
Why should the potential overlap be considerable
in the particular case of the NMOD relation between the two components?

Paragraph 575 to 580 is quite puzzling.
Why does the whole paper make use of higher order dependency features
and then suddenly, at the critical point of actually measuring the crucial
metric
of similarity between composed and observed phrasal vectors, you use
first order features only?

Note 3 is supposed to provide an answer, but I don't understand the explanation
of why the 2nd order paths in the composed representations are not reliable,
please clarify.

Section 6

"Smoothing the PPMI calculation with a value of Î± = 0.75 generally has a 663
small positive effect."
does not seem so obvious from table 3.

What are the optimal values for h and q in equation 8 and 9? They are important
in order to estimate
how much of "hybridity" provides the slight gains with respect to the unaligned
results.

It seems that in table 4 results correspond to using the add combination, it
could help to have this in the legend.
Also, couldn't you provide the results from the word2vec vectors for the
compound phrases?

I don't understand the intuition behind the FREQ baseline. Why would a frequent
compound tend to be compositional? This suggests maybe a bias in the dataset.